# Humoral Immunity to Allogeneic Immunoproteasome-Expressing Mesenchymal Stromal Cells Requires Efferocytosis by Endogenous Phagocytes

**DOI:** 10.3390/cells11040596

**Published:** 2022-02-09

**Authors:** Jean-Pierre Bikorimana, Jamilah Abusarah, Natasha Salame, Nehme El-Hachem, Riam Shammaa, Moutih Rafei

**Affiliations:** 1Department of Microbiology, Infectious Diseases and Immunology, Université de Montréal, Montreal, QC H3T 1A8, Canada; jean.pierre.bikorimana@umontreal.ca; 2Department of Pharmacology and Physiology, Université de Montréal, Montreal, QC H3T 1A8, Canada; jamila_abusara@hotmail.com (J.A.); hachemn@gmail.com (N.E.-H.); 3Department of Biomedical Sciences, Université de Montréal, Montreal, QC H3T 1A8, Canada; salame.natasha@gmail.com; 4Pediatric Hematology-Oncology Division, Centre Hospitalier Universitaire Sainte-Justine Research Centre, Montreal, QC H3T 1C5, Canada; 5Canadian Centers for Regenerative Therapy, Toronto, ON M5R 1A8, Canada; rshamma@intellistemtech.com; 6IntelliStem Technologies Inc., Toronto, ON M5R 3N5, Canada; 7Molecular Biology Program, Université de Montréal, Montreal, QC H3T 1A8, Canada

**Keywords:** mesenchymal stromal cells, immunoproteasome, antigen presentation, antibodies, cytokines, allogeneic, CD47, efferocytosis, cross-priming, tumor growth

## Abstract

The extensive use of mesenchymal stromal cells (MSCs) over the last decade has revolutionized modern medicine. From the delivery of pharmacological proteins to regenerative medicine and immune modulation, these cells have proven to be highly pleiotropic and responsive to their surrounding environment. Nevertheless, their role in promoting inflammation has been fairly limited by the questionable use of interferon-gamma, as this approach has also been proven to enhance the cells’ immune-suppressive abilities. Alternatively, we have previously shown that de novo expression of the immunoproteasome (IPr) complex instills potent antigen cross-presentation capabilities in MSCs. Interestingly, these cells were found to express the major histocompatibility class (MHC) II protein, which prompted us to investigate their ability to stimulate humoral immunity. Using a series of in vivo studies, we found that administration of allogeneic ovalbumin (OVA)-pulsed MSC-IPr cells elicits a moderate antibody titer, which was further enhanced by the combined use of pro-inflammatory cytokines. The generated antibodies were functional as they blocked CD4 T-cell activation following their co-culture with OVA-pulsed MSC-IPr and mitigated E.G7 tumor growth in vivo. The therapeutic potency of MSC-IPr was, however, dependent on efferocytosis, as phagocyte depletion prior to vaccination abrogated MSC-IPr-induced humoral responses while promoting their survival in the host. In contrast, antibody-mediated neutralization of CD47, a potent “do not eat me signal”, enhanced antibody titer levels. These observations highlight the major role played by myeloid cells in supporting antibody production by MSC-IPr and suggest that the immune outcome is dictated by a net balance between efferocytosis-stimulating and -inhibiting signals.

## 1. Introduction

Hosts are continuously exposed to external insults, including invading pathogens [1]. The harmony and interaction between different cellular and acellular components of both arms of the immune system is key for achieving the complete, successful eradication of these insults. By working in tandem, the innate arm of the immune system provides a quick and early response, albeit lacking the ability to generate specific long-term memory [2,3]. The adaptive immune system, on the other hand, provides a more specific immune response towards a given antigen and is associated with long-term memory. The latter point is particularly important as it enables the body to respond quickly and in a more effective way in case of a recall years after the initial encounter [3].

One of the central components of adaptive immunity is the production of antibodies (immunoglobulins) by plasma cells. This “Y” shaped protein binds to a specific epitope, leading to either pathogen/pathogen-derived particle neutralization or activation of the complement system [4]. Moreover, antibodies can form immune complexes when bound to their cognate antigens, consequently resulting in several immune cascades upon their interaction with the Fc receptor (FcR) on the surface of certain immune cells. As a result, antibody-dependent cellular cytotoxicity (ADCC), pathogen or particulate clearance by phagocytosis, modulation of inflammatory responses or cytokine release can occur [5,6,7,8,9]. Thus, humoral immunity can provide protective responses through a wide range of mechanisms.

For an effective antibody response to be induced, a vaccine needs to elicit a T-helper response (e.g., activation of a specific CD4 T-cell subset) [10]. This requires proper antigen processing and presentation by specialized antigen-presenting cells (APCs) such as dendritic cells (DCs) [11]. Given the importance of DCs in priming adaptive immunity, the field of antigen presentation mainly focuses on: (i) optimizing DC antigen presentation and function, (ii) developing strategies to polarize DCs into specialized subtypes, and/or (iii) targeting antigens to specific cell surface receptors expressed on DC cell surfaces [12,13,14]. Despite years of extensive research, the use of DCs remains challenging due to major manufacturing hurdles and difficulties in the ex vivo generation of a high number of functional DCs capable of yielding meaningful clinical outcomes following their administration to patients [15]. In fact, only one DC cancer vaccine (Sipuleucel-T) has been approved so far by the FDA for the treatment of metastatic castrate-resistant hormone-refractory prostate cancer [16]. Despite the initial encouraging data, this vaccine failed to grant a long-term cellular response and led to the production of low to medium anti-PAP antibody titers [17,18]. Thus, additional work is warranted for the development of cellular vaccines capable of triggering potent and long-lasting immunity.

An elegant study by Abusarah et al. recently reported a newly engineered cell vaccine designed to overcome most of the hurdles seen with the use of DCs in cancer vaccination [19]. By de novo expression of the immunoproteasome (IPr) complex in mesenchymal stromal cells (MSCs), potent activation of the immune system was achieved [19]. More specifically, the use of a both syngeneic and allogeneic MSC-IPr vaccine pulsed with tumor cell lysate cured pre-established T-/B-cell lymphomas as well as B16F0 melanoma. The therapeutic effect was further pronounced when combined with the immune-checkpoint inhibitor PD-1 and the agonist antibody targeting 4-1BB. Interestingly however, transcriptomic analysis revealed that MSC-IPr can mediate MHCII antigen presentation—hence their capacity to elicit humoral responses. We thus tested this hypothesis and provide evidence that these cells can indeed trigger an antibody response effective enough to impair cancer growth in vivo. The study also highlights possible optimization steps using two pro-inflammatory cytokines, as an example, to enhance the outcome of MSC-IPr-induced humoral responses.

## 2. Materials and Methods

### 2.1. Animals and Ethics

The mice used in the study; C57BL/6, Balb/c, and OT-II strains, were purchased from Jackson Laboratories (Bar Harbor, ME, USA) and housed in a pathogen-free environment at the animal facility of the Institute for Research in Immunology and Cancer (IRIC). For all experiments, female mice were used at 6–8 weeks old. The Animal Ethics Committee of Université de Montréal approved detailed experimental procedures.

### 2.2. Cell Lines and Reagents

The E.G7 T-cell lymphoma line was kindly provided by Dr. Jacques Galipeau (University of Wisconsin–Madison, WI, USA). All cell culture media and reagents were purchased from Wisent Bioproducts (St Jean-Baptiste, QC, Canada). The antibodies used in the flow cytometry, including I-Ab and CD47 antibodies, were purchased from BD Biosciences (San Jose, CA, USA). The IL-2 Quantikine kit was purchased from R&D Systems (Minneapolis, MN, USA). The chicken egg white ovalbumin (OVA) protein was purchased from Sigma-Aldrich (St-Louis, MI, USA). Recombinant murine IFNγ, GM-CSF and IL-21 were purchased from Peprotech (Rocky Hill, NJ, USA). The SIINFEKL and ISQAVHAAHAEINEAGR peptides were synthesized by Genscript (Piscataway, NJ, USA). The CD4 T-cell isolation kit was purchased from StemCell Technologies (Vancouver, BC, Canada). Liposomes and liposome-clodronate were purchased from Liposoma Research (Amsterdam, The Netherlands). XenoLight D-Luciferin K+ Salt was purchased from PerkinElmer (MA, USA). An Annexin-V staining kit was purchased from Cedarlane (Burlington, ON, Canada).

### 2.3. Generation of MSC-IPr Cells

To generate bone marrow (BM)-derived MSCs, the femurs of 6–8 weeks old female C57BL/6 mice were flushed with Alpha Modification of Eagle’s Medium (AMEM) supplemented with 10% FBS, and 50 U/mL Penicillin–Streptomycin in a 10 cm^2^ cell culture dish. After 48 h, non-adherent cells were removed. The media was changed every 3 to 4 days. When the cells reached 80% confluency, adherent cells were detached using 0.25% Trypsin, harvested and expanded until a homogenous population was obtained. The collected cells were validated using flow cytometry for the expression of the surface markers CD44, CD45, CD73, CD90 and CD105. The osteogenic and adipogenic differentiation capacity of the resulting MSCs was tested as previously described [20]. Generated MSCs were expanded and frozen at passage number 9 or 10 to be used in future experiments.

To generate the MSC-IPr from BM-derived MSCs, a retroviral construct designed to deliver the main three subunits of the immunoproteasome (IPr) was used to transduce MSCs, as previously described in detail elsewhere [19]. The transduction efficiency was confirmed by assessing eGFP expression using flow cytometry and immunoblotting of the IPr subunits. Similar steps were followed using the AP2 construct backbone to generate control (Ctl) MSCs.

### 2.4. Antigen Presentation Assay

To assess the ability of the MSC-IPr to present a specific antigen, cells were seeded at 25 × 10^3^ cells per well in a 24-well plate (Corning). The following day, the plated cells were washed and pulsed with media containing 1 μg/mL of the SIINFEKL or ISQAVHAAHAEINEAG peptide for 9 h. The technical control OVA antigen was used at a concentration of 5 mg/mL. At the end of the pulsing period, the media was removed, and the cells washed to remove traces of antigen prior to their co-culture with 10^6^/^mL^ CD4 T-cells purified from the spleen of OT-II mice, using the CD4 positive isolation kit according to the manufacturer’s protocol. The supernatants were collected three days later to quantify IL-2 levels by ELISA, according to the manufacturer’s protocol.

To evaluate the efficacy of the antibodies produced by the immunized mice in binding to OVA antigens and the influence this binding might have on the CD4 T cell immune response, two similar antigen presentation assays were conducted, with modifications. Sera from mice immunized by receiving two IP injections of OVA-pulsed MSC-IPr on days 0 and 14 were used as a source of OVA-specific antibodies (collected at week 6 post-second dosing). For the modified protocol, the collected sera were admixed with OVA (1:100) 1 h prior to MSC-IPr pulsing, or 1 h prior to adding OT-II-derived CD4 T cells onto OVA-pulsed MSC-IPr. Three days later, supernatants were collected, and the level of IL-2 was quantified using ELISA, according to manufacturer’s protocol.

### 2.5. RNA Sequencing and Bioinformatics Analysis

The collection, isolation and sequencing of cellular RNA were performed as previously described [19]. Raw RNA-seq read count data were summarized at the gene level with Htseq-count (version 0.10 add ref) from reads aligned to the mouse genome (GRCm39). Differentially expressed genes between MSC-IPr and Ctl MSCs were calculated by DESeq2 (v.1.26) [19]. Pre-ranked gene set enrichment (GSEA) was performed on the list of differentially expressed genes according to their log2 fold change. Custom R scripts were used to filter highly redundant biological processes by calculating the Jaccard Coefficient between members of any given pairs of GO terms and keeping the term with the lowest p-value from GSEA. A false discovery rate (FDR) of 0.05 was considered as an acceptable threshold for further investigation of differentially regulated genes. All analyses were conducted in R (v3.6.1). The ggplot2, ClusterProfiler and dplyr packages were used for enrichment analyses and heatmap generation.

### 2.6. Immunization Studies

To evaluate the outcome of different immunization approaches in vivo, a dose response study was conducted by immunizing 6–8 week-old female Balb/c mice (*n* = 5/group) using 1 × 10^6^ C57BL/6-derived MSC-IPr pulsed for 9 h with OVA (5 mg/mL) versus unpulsed MSC-IPr. The mice were immunized using the intraperitoneal (IP) route at days 0 and 14. Blood samples were collected every two weeks starting at day 21 post-initial dose to obtain the sera. Following the same protocol, a similar immunization study was then conducted to compare the IP versus the subcutaneous (SC) route using 1 × 10^6^ MSC-IPr cells (for this study, blood samples were collected at week 6 post-initial immunization dose to obtain the sera). For studies evaluating the impact of cytokine co-administration on the humoral response, GM-CSF (50 μg/kg), IL-21 (50 μg/kg) or both cytokines combined were administered at days 0, 2, and 4 and then at days 14, 16 and 18 via the IP route.

To block the efferocytosis of MSC-IPr by phagocytes after injection, 6–8 week-old female C57BL/6 mice *(n* = 6/group) were IP-injected with liposome-clodronate or control liposome (50 μL/20 g mouse) one day prior to immunization using 1 × 10^6^ OVA-pulsed MSC-IPr cells, delivered SC. Two weeks later, the same process was repeated prior to the second immunization dose. For CD47 neutralization in vivo, 1 × 10^6^ MSC-IPr were first admixed with anti-CD47 for 30 min prior to their administration in vivo at days 0 and 14. Sera were collected at week 6 post-initial dosing to quantify the levels of blood IgG.

### 2.7. Antibody Titer Analysis by ELISA

To identify the titer of antibody production at specific time points post-immunization, a volume of 100 μL at 1 μg/mL of OVA protein was diluted in PBS and used for coating a 96-well microtiter Nunc MaxiSorp plates (Thermo Fisher Scientific; Cat#: 442404). The plates were then incubated overnight at 4 °C. The following day, the coating solution was discarded, and the plates were blocked with 150 μL of PBS containing 3% skim milk for 1 h at room temperature. In parallel, sera samples were diluted 1:100, followed by two-fold dilutions in PBS containing 1.5% skim milk and 0.05% Tween using U-bottom microplates or Eppendorf tubes. Once the sera dilutions were ready, the blocking buffer was discarded and all the plates were washed using 400 μL/well of PBS containing 0.05% Tween-20. The wash step was repeated three times. Then, 100 μL of diluted sera test samples were added to each well, followed by 2 h incubation at room temperature. At the end of incubation period, the samples were discarded and the wells were washed three times as mentioned above, followed by the addition of 100 μL/well of anti-mouse IgG conjugated HRP antibody (R&D systems; cat#: HAF007) diluted 1:1000 in PBS containing 1.5% skim milk and 0.05% Tween. The plates were incubated with the detection antibody for 2 h at room temperature. Finally, all wells were washed three times, as mentioned earlier, and the signal was detected by adding 100 μL of TMB (3, 3′, 5, 5′-Tetramethylbenzidine; Sigma; Cat#: T0440-1L). At the end of the incubation period (20 min), 50 µL of 1 M H_2_S0_4_ is added to stop the reaction. Optical density was measured using a microplate reader set to 450 nm and 570 nm wavelengths. The results were presented as end-titers with an optical density higher than the background line set according to the optical density obtained with the pre-immune sera.

### 2.8. Cytokine and Chemokine Analyses

To assess the influence of cytokine co-administration, recombinant GM-CSF, IL-21 or their combination were administered to animals along with OVA-pulsed MSC-IPr injection. Cytokines were administered every 48 h (three injections at week 1) and the same cycle was repeated at week 3 for a total of 6 injections. Sera were collected weekly, whereas cytokine analysis was conducted at the end of the experiment (week 16). The obtained CD4 T-cells were then co-cultured for 72 h with MSC-IPr previously pulsed with ISQAVHAAHAEINEAGR (1 μg/mL). Supernatants were then collected and analyzed by Luminex at Eve Technologies (Calgary, AB, CA) to evaluate the cytokine and chemokine secretion profile of the responding CD4 T-cells.

### 2.9. Assessment of Cell Persistence Post-Injection

The live in vivo imaging study was designed to evaluate the persistence of MSC-IPr cells in vivo after a SC injection in the presence or absence of efferocytosis. For this experiment, MSC-IPr were transduced to stably express firefly luciferase and kept under selection pressure using puromycin. One day before immunization, C57BL/6 female mice (*n* = 6/group), received an injection containing either clodronate liposome or control liposome IP (50 μL/20g mouse). On the following day, the mice were injected SC with 1 × 10^6^ MSC-IPr. Bioluminescence signals were recorded at days 0 (3 h after injection), 3 and 7 post-injection. For each imaging session performed at the IRIC (Montreal, QC), mice received an IP injection of 0.2 mL of 15 mg/mL XenoLight D-Luciferin—K+ Salt (equivalent to 30 mg/kg). Mice were kept under 1.5–2.5% inhaled isoflurane anesthesia and the bioluminescence signal was acquired after 10 min using the Prism in vivo imaging system (Médilumine, QC, Canada). The acquired data were then plotted as luciferase signal decay.

### 2.10. Tumor Neutralization Studies

To assess the therapeutic abilities of antibodies generated by immunized mice, E.G7 cells were admixed with control or OVA-specific sera collected from immunized mice at a 1:100 dilution for 20 min at 37 °C. Following cell washing by PBS to remove unbound antibodies, 5 × 10^5^ cells were injected via the SC route in C57BL/6 mice. Tumor growth was followed every 4 days until reaching the endpoints.

### 2.11. Generation of Heat-Killed (HK) MSCs

Ctl MSCs and MSC-IPr were heat-killed as previously described [21]. Briefly, MSCs were re-suspended in sterile PBS then incubated in a water bath set at 50 °C for 15–20 min. Cell death was then confirmed by conducting an annexin-V and analyzed by flow cytometry.

### 2.12. Statistical Analysis

*P*-values were calculated using both the one-way analysis of variance (ANOVA) and the Student’s *t* test, where applicable. Statistical significance is represented as follows: * *p <* 0.05, ** *p* < 0.01, *** *p* < 0.001. Statistical tests used for bioinformatic analysis are described in their corresponding section.

## 3. Results

### 3.1. MSC-IPr Can Effectively Present MHCII Peptides to CD4 T Cells

The generation and full characterization of MSC-IPr cells has been previously reported [19]. Briefly, MSC-IPr cells were engineered by transducing wild-type (WT) MSCs with retroviruses produced with a plasmid construct harboring the three main subunits of the murine IPr—cloned in tandem repeats to ensure their equimolar expression (Figure 1A). Although WT MSCs modified with the empty vector (Ctl MSCs) remained I-A^b^ negative, both MSC-IPr and MSCs treated with IFN-gamma (MSCγ) had comparable I-A^b^ expression levels, as detected by flow cytometry (Figure 1B,C). To functionally validate this observation, an antigen presentation assay was conducted using OT-II-derived CD4 T cells specific to the OVA-derived peptide 323-339 presented by I-A^b^ molecules (Figure 1D). As anticipated, pulsing of MSC-IPr with the OVA protein or the I-A^b^ OVA-derived peptide 323–339 triggered IL-2 production from responding CD4 T cells, in contrast to the absent signal obtained using the OVA-derived SIINFEKL peptide that is normally recognized by CD8 T cells (Figure 1E). These observations correlate with the differentially expressed genes detected by RNA-seq (Ctl MSCs versus MSC-IPr), where 6.75% (51 genes) of observed changes were associated with MHCII antigen processing and presentation (Figure 1F). Some of the upregulated genes include several kinesin motor proteins (*Kif2c*, *Kif4a*, *Kif4b*, *KIF5a*, *Kif11*, *Kif15*, *Kif18*, *Kif22*, and *Kifap3*), the RAB-interacting lysosomal protein (*Rilp*), Legumain (*Lgmn*), H2-DMB2 (involved in MHCII complex assembly), and the Rac GTPase-activating protein 1 (*Racgap1*). An additional gene set enrichment analysis was performed on the ranked list of MSCs versus MSC-IPr differentially expressed genes that identified MHCII antigen processing and presentation (GO biological process) as being significantly upregulated (NES = 1.65; *p* < 0.01) in Ctl MSCs (Figure 1G). Significantly, upregulated genes were ranked by their corresponding log2 fold change and included several kinesin motor proteins (Figure 1H). In summary, MSC-IPr exhibit an interesting ability to present peptides via their MHCII molecules in addition to excelling at antigen cross-presentation, as previously reported [19].

### 3.2. Humoral Responses Induced by Allogeneic MSC-IPr Are Both Dose- and Route-Dependent

The ability of MSC-IPr to activate CD4 T cells prompted us to investigate whether these cells can be used to stimulate antibody production in vivo. To test this hypothesis, OVA-pulsed C57BL/6-derived MSC-IPr were used to immunize allogeneic Balb/c mice using three different cellular doses (0.1, 0.5 or 1 × 10^6^ cells—Figure 2A), delivered via an IP injection. Analysis of sera from immunized animals revealed good OVA-specific antibody titer with a sustainable response detected up to eight weeks using the highest MSC-IPr dose (1 × 10^6^—Figure 2B). Injection of lower MSC-IPr doses, on the other hand, led to a noticeable antibody titer decrease by week 8 post-initial dosing. Since the route of vaccination can significantly affect the therapeutic potency of the vaccine [22], we next compared IP and SC vaccination routes (Figure 2C), and found that the latter route led to significantly higher antibody titers when analyzed at week 4 post-initial dosing (Figure 2D). The sum of these data infers that the humoral response to OVA-pulsed MSC-IPr is both dose- and route-dependent.

### 3.3. Co-Administration of Pro-Inflammatory Cytokines Enhances Humoral Responses

With a moderate antibody response generated by SC-injection of allogeneic MSC-IPr, we next investigated whether the potency of the vaccine could be enhanced when delivered in conjugation with two pro-inflammatory cytokines. We selected granulocyte-macrophage colony-stimulating factor (GM-CSF) due to its role in supporting the differentiation of monocytes and dendritic cell precursors while stimulating their antigen presentation ability [23]. As a second cytokine we chose interleukin (IL)-21, as it acts as a B-cell stimulating factor and supports IgA isotype switching [24,25]. When co-injected with GM-CSF or IL-21, OVA-pulsed allogeneic MSC-IPr led to an overall antibody response comparable to the standalone MSC-IPr group (Figure 3A). Interestingly, however, the combined use of the GM-CSF and IL-21 during vaccination led to a higher antibody titer that remained sustainable over 14 weeks compared to the other treatment groups (Figure 3A). When the humoral response was further assessed to identify Ig isotypes, IgG_3_ and IgE were undetected, whereas IgG_1_ was predominantly induced in all groups, followed by both IgG_2a_ and IgG_2b_ antibodies (Figure 3B). Although lower in magnitude, IgA antibodies could only be detected in the IL-21- and GM-CSF/IL-21-treated groups (Figure 3B). To further characterize the induced humoral response, CD4 T cells isolated from immunized mice were co-cultured with ISQAVHAAHAEINEAGR-pulsed MSC-IPr. Luminex analysis of conditioned media revealed differential cytokine/chemokine production by responding CD4 T cells. While IFNγ was commonly produced by all groups (Figure 3C), co-injection of MSC-IPr with GM-CSF triggered the production of IL-2, IL-3, macrophage inflammatory protein (MIP)1α and MIP2, whereas the use of IL-21 diminished chemokine levels at the expense of IL-2 and IL-3 secretion (Figure 3C). Interestingly, the GM-CSF and IL-21 combination induced the production of all of the above-mentioned factors with enhanced monocyte chemoattractant protein (MCP)-1 and MIP1β production (Figure 3C). Altogether, our data allude to a model in which humoral responses triggered by allogeneic MSC-IPr vaccination can be further improved via the co-administration of GM-CSF and IL-21.

### 3.4. Generated Antibodies Neutralize OVA Peptide-MHCII Complexes but Enhance OVA-Antibody Complex Uptake by MSC-IPr

Antibodies play crucial immune-related roles by mediating antigen neutralization, and/or facilitating target opsonization as a means to activate the complement system and/or drive FcγR-mediated phagocytosis [4]. Thus, we next validated the in vitro functionality of our triggered antibodies using two OT-II-based antigen presentation assays. In the first assay, MSC-IPr were first pulsed with the OVA protein prior to adding sera collected from mice immunized with OVA-pulsed MSC-IPr, 1 h prior to their co-culture with CD4 T cells isolated from the spleen of OT-II mice (Figure 4A). When assessed for IL-2 production, OT-II activation was impaired in the group containing OVA-specific sera (Figure 4B). On the other hand, when OVA was admixed with sera for 1 h prior to the antigen-loading step (Figure 4C), IL-2 production was enhanced compared to the OVA group treated with pre-immune sera (Figure 4D). These results clearly indicate that the humoral response induced following allogeneic OVA-pulsed MSC-IPr immunization can generate antibodies capable of binding and blocking peptide-MHCII complexes or form immune complexes with the OVA protein, leading to enhanced uptake by target APCs.

### 3.5. MSC-IPr-Generated Antibodies Can Impair Tumor Growth In Vivo

In light of the data obtained in vitro, we next evaluated whether the generated sera contained antibodies capable of binding peptides presented on the cell surface of E.G7 tumor cells (Figure 5A). Although E.G7 cells are negative for I-A^b^ expression (Figure 5B), flow cytometry analysis revealed that the addition of immunization-generated sera cross-reacts partially with the E.G7 cell surface, in contrast to antibodies found in pre-immune sera (Figure 5C). Based on these observations, we next designed an in vivo experiment in which E.G7 tumor cells were first mixed with pre-immune or OVA-specific sera collected from MSC-IPr immunized mice prior to syngeneic tumor transplantation (Figure 5A,D). A significant delay in tumor growth was observed in the group transplanted with E.G7 tumor cells mixed with OVA-specific antibodies (Figure 5E), clearly indicating a beneficial therapeutic effect mediated by the MSC-IPr-induced humoral response.

### 3.6. Humoral Responses Triggered in Response to Allogeneic MSC-IPr Require Efferocytosis

MSCs are widely reported to influence the response of the immune system via multiple mechanisms including the secretion of soluble factors, cellular interactions, or released exosomes [26,27,28]. More recent reports have revealed another mechanism of action for MSCs via efferocytosis by endogenous phagocytes under pro-inflammatory conditions [29,30]. In addition, several cell surface markers have been reported to function as “eat-me or do not eat-me signals” capable of inducing or blocking efferocytosis, respectively (Figure 6A) [31,32]. Since efferocytosis could play a major role in enhancing humoral immunity to allogeneic OVA-pulsed MSC-IPr, we first assessed the expression of CD47, a cell surface protein capable of blocking efferocytosis by binding to signal regulatory protein (SIRP)α on phagocytes [33,34], on the surface of MSC-IPr and Ctl MSCs. Although both cell types were positive for this marker (Figure 6B), the intensity of CD47 expression was significantly higher on the surface of MSC-IPr (Figure 6C). To investigate whether efferocytosis blockade by CD47 neutralization ultimately plays a negative role on the elicited humoral response, we next designed an in vivo experiment where anti-CD47 antibodies were injected to animals undergoing allogeneic MSC-IPr immunization. In parallel, a second group of mice received clodronate injections (to deplete phagocytes) a day prior to immunization, to further assess the importance of efferocytosis (Figure 6D). As shown by the antibody titer analysis, phagocyte depletion by clodronate abrogated antibody production, whereas CD47 neutralization improved IgG titers (Figure 6E). These results prompted us to further investigate the importance of efferocytosis on the humoral response induced by allogeneic MSC-IPr immunization by assessing their persistence in vivo following their SC delivery (Figure 6F). Live in vivo imaging revealed that clodronate injection improves MSC-IPr survival in allogeneic mice up to 7 days post-injection, whereas animals treated with control liposome cleared injected cells within a 3-day period (Figure 6G,H). Phagocyte-mediated efferocytosis in this context could not be attributed to enhanced phosphatidylserine levels (a signal for phagocyte-mediated clearance) as live MSC-IPr exhibits weaker annexin-V staining at steady states compared to Ctl MSCs (Figure 6I–K). Altogether, these observations indicate that allogeneic MSC-IPr can elicit antibody production but require endogenous phagocyte-mediated efferocytosis to do so.

## 4. Discussion

MSCs are generally used in regenerative medicine or as an immune-modulating therapy for the treatment of several autoimmune diseases or catastrophic illnesses such as graft-versus-host disease [35,36,37,38,39]. However, our group has demonstrated that MSCs can be converted to potent APCs following pharmacological treatment with the agonist pyrimido-indole derivative UM171a or by modulating the proteasomal machinery [19,21,40]. One example of the latter approach consists of engineering MSCs to express the IPr complex. These cells not only excel at antigen cross-presentation, but can also elicit powerful anti-tumoral responses capable of eradicating established lymphoma and melanoma tumors [19]. Interestingly, however, transcriptomic analysis of MSC-IPr has clearly revealed their ability to mediate antigen presentation via MHCII, which was confirmed in this study.

In a nutshell, antigen-pulsed allogenic MSC-IPr are effective at mounting humoral responses. Although the antibody titer was moderate, it could be further enhanced via the use of pro-inflammatory cytokines such as GM-CSF and IL-21, amongst many others. This combination is interesting, as MSC-IPr co-administration with each cytokine delivered individually led to a response comparable to the cellular vaccine as a stand-alone therapy, whereas GM-CSF and IL-21 co-administration with the vaccine enhanced and sustained antibody titers. These data allude to synergistic effects, which could potentially be further enhanced with the use of the GIFT-21 fusokine (GM-CSF fused to IL-21). In fact, this fusion protein was shown to promote monocyte maturation into a distinct hyperactivated DC population displaying enhanced antigen presentation properties [41,42]. Future studies can perhaps compare the outcomes derived from combining the MSC-IPr vaccine with GM-CSF/IL-21 versus the GIFT-21 fusokine to provide a more solid vaccination regimen. Nevertheless, pre-treatment of the OVA-expressing E.G7 cell line with the generated antibodies prior to their transplantation significantly delayed tumor growth in vivo. Although beyond the scope of this study, one can stipulate that the observed tumor delay is driven by complement-dependent cytotoxicity, antibody-dependent phagocytosis and/or antibody-dependent cellular cytotoxicity [4,5]. Altogether, these observations are in line with our previous study demonstrating that MSC-IPr-induced anti-tumoral immunity requires both CD8 and CD4 T cells (mostly likely participating in antibody production) to protect the host from cancer growth [19].

In terms of the underlying mechanism, the current consensus is that MSCs mediate their suppressive effect via the production of soluble mediators or exosome release [35,36,37,38,39]. However, the new school of thought stipulates that MSCs must undergo apoptosis and/or stimulate efferocytosis by endogenous phagocytes prior to exerting their immune suppressive abilities in vivo [29,30]. This outcome is not different in our case, as clodronate-mediated depletion of phagocytes prior to immunisation prolongs allogeneic MSC-IPr survival in vivo while blunting antibody production. On the other hand, inhibiting the “do not eat-me” signal CD47, an integrin that interacts with the inhibitory transmembrane receptor SIRPα present on myeloid cells, results in significantly higher antibody titers. These results suggest an important interplay occurring between myeloid cells and MSCs, which, if further enhanced, could lead to a substantial boost to humoral immunity. Nevertheless, efferocytosis by myeloid cells is not limited to monocytes/macrophages, as cDC1 and cDC2 were also shown to capture dying MSCs [30]. What is the exact nature of phagocytic cells mediating MSC-IPr efferocytosis? Are MSC-IPr being cleared by efferocytosis via specific signals or following cell death by apoptosis following their in vivo administration? Can efferocytosis-mediated uptake by a given myeloid cell be specifically targeted or enhanced? These are some of the questions that need to be further addressed experimentally.

In sum, this proof-of-concept study demonstrates that MSC-IPr can capture, process and present antigens to responding CD4 T cells, but require endogenous efferocytosis to cross-prime phagocytes in order to elicit antibody production by differentiating plasma cells. Although we are just starting to understand the MSC-IPr mode of action, these encouraging first steps imply that allogeneic MSC-IPr cells can be indeed exploited as cellular vaccines to induce both humoral and cytotoxic immunity, as previously reported [19,22]. Further mechanistic investigations are however required to better understand how the IPr complex (strictly involved in MHCI peptide generation) facilitates the supply of MHCII-restricted peptides in MSCs, and whether such an effect is related to the absent assembly of the three IPr subunits within the mature 20S or 26S proteasome complex.

## 5. Patents

The authors declare that they have no competing financial interests. A provisional patent has been filed to protect the MSC-IPr technology and its applications (#62/835,678).

## Figures and Tables

**Figure 1 cells-11-00596-f001:**
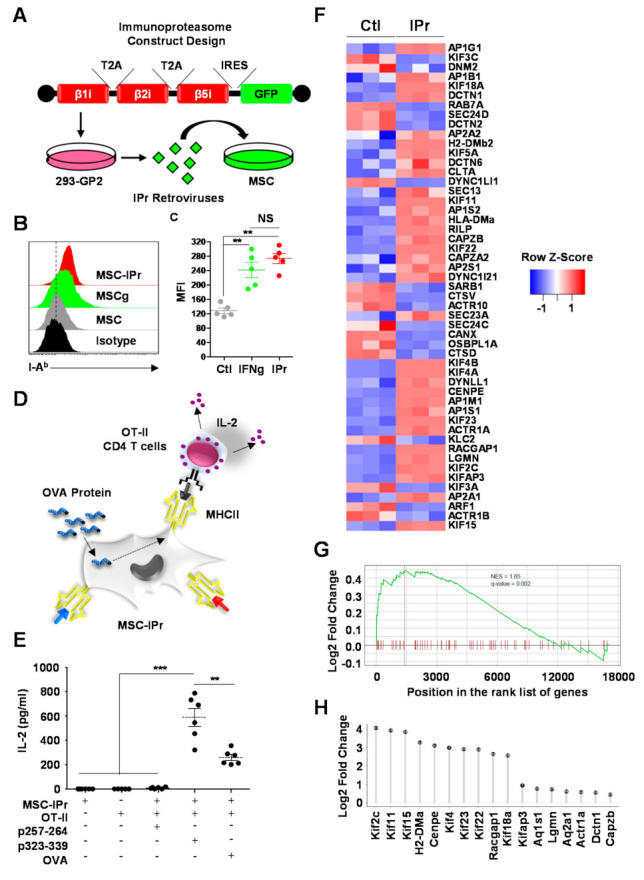
Characterizing the antigen presentation capacity of antigen-pulsed MSC-IPr to CD4 T cells. (**A**) Representative schematic diagram showing MSC-IPr engineering. (**B**,**C**) Representative flow cytometry assessment of MHCII expression (**B**) and its mean fluorescent intensity (MFI-C). Ctl MSCs are shown in grey, MSCγ in green and MSC-IPr in red. Isotype staining is shown in black. (**D**) Schematic diagram showing the antigen presentation assay used to evaluate the CD4 T-cell response. (**E**) ELISA quantification of IL-2 produced in response to OVA- or OVA peptide-pulsed MSC-IPr co-cultured with OT-II-derived CD4 T cells. (**F**) The heatmap represents the z-scored expression level of the differentially expressed genes from the MHCII antigen presentation biological process in Ctl MSCs versus MSC-IPr. Upregulated and downregulated genes are highlighted in red and blue respectively. (**G**) This plot shows the running enrichment score from the pre-ranked GSEA of the ANTIGEN_VIA_MHC_CLASS_II GO process. NES is the normalized enrichment score and the q-value threshold tests for a false discovery rate of 5% among all the significant GO terms. (**H**) The plot illustrates all ANTIGEN_VIA_MHC_CLASS_II genes contributing to the maximum enrichment score. Corresponding log2 fold change is shown in the Y-axis dots. For panel C, *n* = 5/group with ** *p* < 0.01. For panel E, *n* = 6/group with ** *p* < 0.01 and *** *p* < 0.001.

**Figure 2 cells-11-00596-f002:**
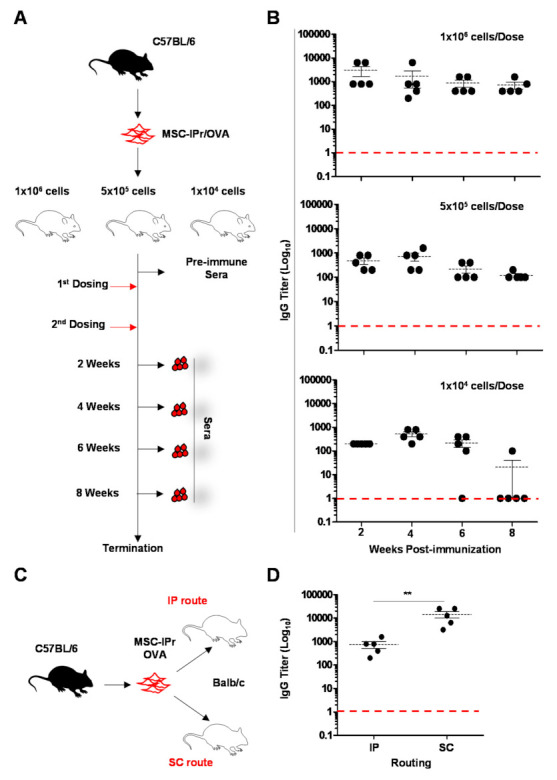
Antibody titers in response to various dosing and routes. (**A**) Schematic diagram representing the immunization approach used to compare the potency of various cellular doses via the IP route. (**B**) Antibody titer analysis conducted every two weeks in response to three different allogeneic MSC-IPr immunization doses. The red dashed line represents the values of pre-immune sera. (**C**) Schematic diagram representing the immunization approach used to compare the IP and SC routes. (**D**) Antibody titer analysis conducted at week 4 post-initial SC dosing using 1 × 10^6^ OVA-pulsed MSC-IPr. The red dashed line represents the values of pre-immune sera. For panels B and D, *n* = 5/group with and ** *p* < 0.01.

**Figure 3 cells-11-00596-f003:**
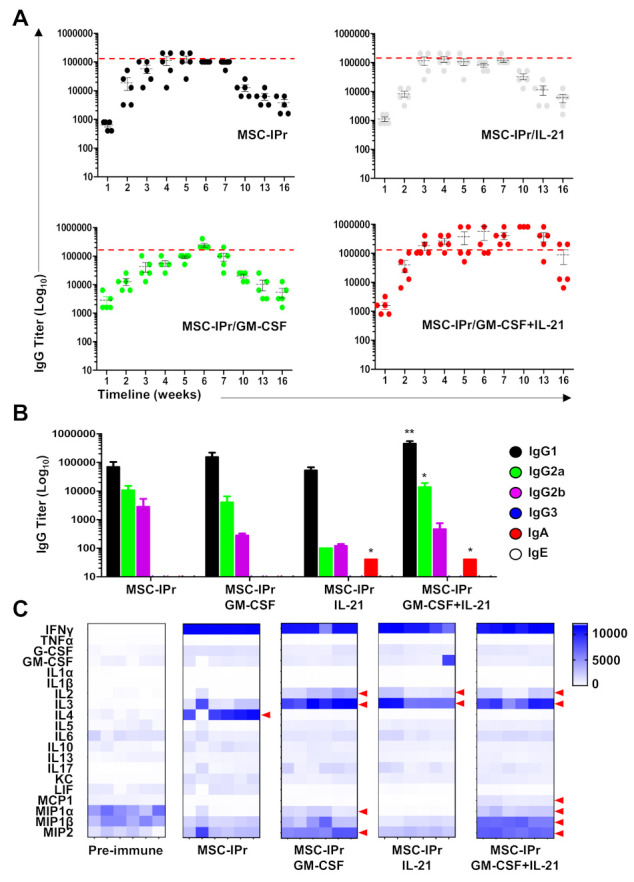
GM-CSF and IL-21 combination improves the immune response to allogeneic MSC-IPr. (**A**) Weekly antibody titer analysis by ELISA using sera collected from allogeneic OVA-pulsed MSC-IPr alone (black) or in combination with IL-21 (white), GM-CSF (green) or a mix of GM-CSF and IL-21 (red). The red dashed line represents the highest titer obtained with MSC-IPr alone. (**B**) Assessing antibody isotypes at week 6 in all test groups. (**C**) Multiplex cytokine analysis of in vitro-restimulated CD4 T cells cultured with ISQAVHAAHAEINEAGR-pulsed MSC-IPr. Red arrows represent detected changes in cytokine/chemokine levels compared to control groups. For panels (**A**–**C**), *n* = 5–6/group with * *p* < 0.05 and ** *p* < 0.01.

**Figure 4 cells-11-00596-f004:**
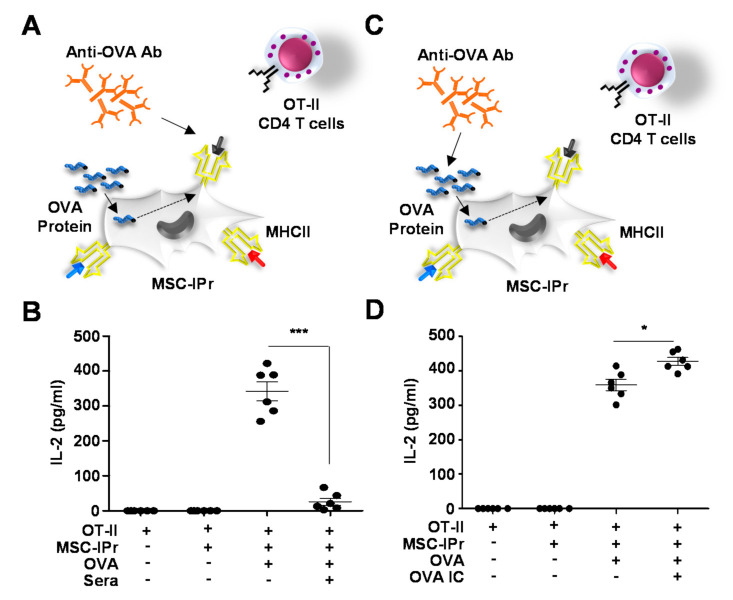
The neutralizing effects of generated sera on in vitro antigen presentation. (**A**) Schematic representation of the antigen presentation assay conducted in the presence of antibodies from immunized mice. Antibodies were added 1 h following OVA pulsing. (**B**) Quantification of IL-2 produced by OTII-derived CD4 T cells during the antigen presentation assay shown in panel (**A**). (**C**) Schematic representation of the antigen presentation assay conducted using OVA admixed with antibodies generated by immunized mice prior to MSC-IPr pulsing. (**D**) Quantification of IL-2 produced by OTII-derived CD4 T cells during the antigen presentation assay shown in panel (**C**). For panels (**B**,**D**), *n* = 6/group with * *p* < 0.05 and *** *p* < 0.001.

**Figure 5 cells-11-00596-f005:**
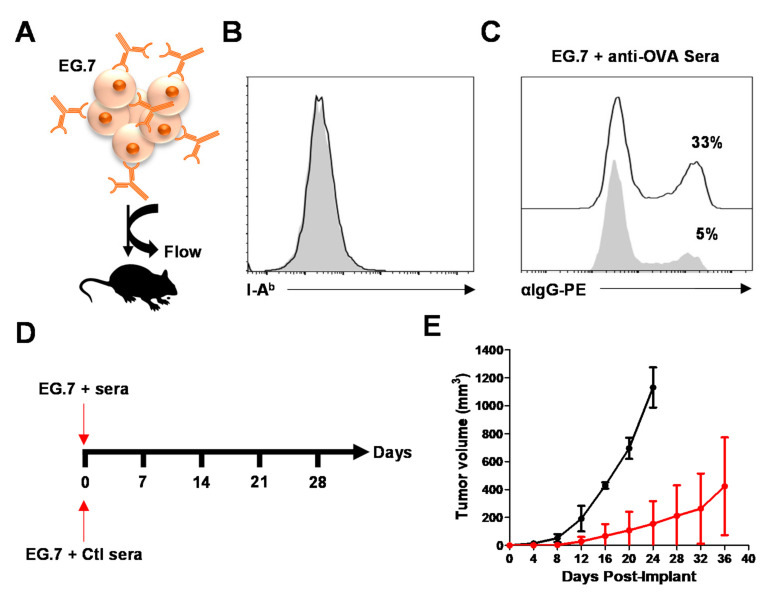
Antibodies bound to E.G7 tumor cells inhibit their growth in vivo. (**A**) Schematic representation of the study design. (**B**) Representative flow cytometry analysis of E.G7 I-A^b^ levels. (**C**) Representative flow cytometry analysis of E.G7 cell surface-bound antibodies. Treatment of E.G7 cells with the pre-immune sera is shown in light grey whereas OVA-specific sera cells are in plain white. (**D**,**E**) Schematic diagram (**D**) of the in vivo experiment conducted using E.G7 tumor cells incubated with antibodies generated by mice immunized using OVA-pulsed MSC-IPr before being transplanted SC in immunocompetent C57BL/6 mice (**E**). E.G7 treated with Ctl sera is shown in black whereas E.G7 mixed with sera isolated from mice immunized using OVA-pulsed allogeneic MSC-IPr is depicted in red. For panel D, *n* = 10/group.

**Figure 6 cells-11-00596-f006:**
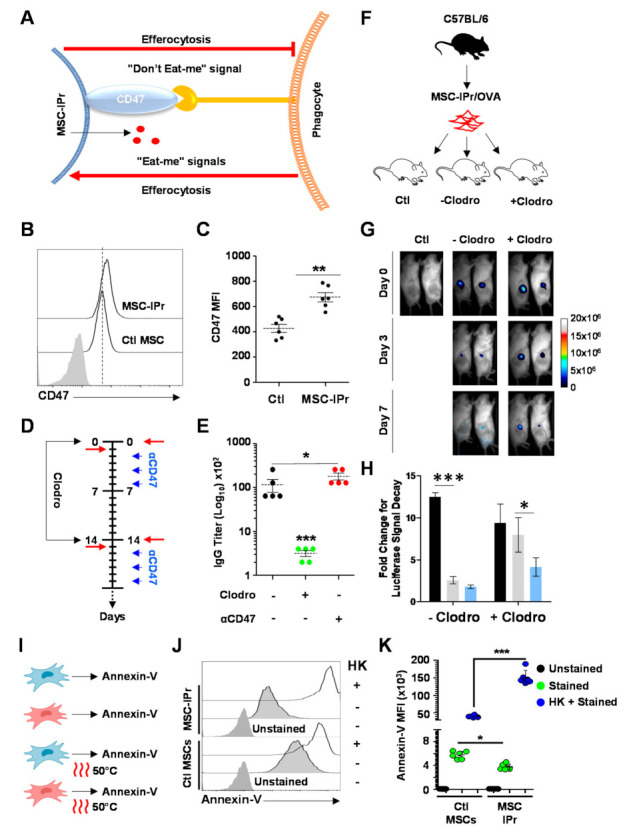
Efferocytosis is required to humoral responses induced by allogeneic MSC-IPr. (**A**) Schematic diagram representing the interplay between MSC-IPr and phagocytes. (**B**,**C**) Representative flow cytometry (**B**) analysis of CD47 expression on the surface of MSC-IPr versus Ctl MSCs and its cognate MFI (**C**). (**D**) Schematic diagram reflecting the immunization schedule OVA-pulsed MSC-IPr in the presence or absence of anti-CD47 or clodronate. Red arrows represent SC injection of MSC-IPr. (**E**) IgG titer analysis by ELISA in sera isolated from mice vaccinated using allogeneic MSC-IPr as a combination with anti-CD47 or clodronate or alone. (**F**) Schematic diagram showing the design of the live in vivo imaging study. (**G**) Representative live in vivo imaging of mice implanted with luciferase-expressing MSC-IPr administered to mice with previous liposome or clodronate-liposome injection. (**H**) Quantification of luciferase luminescence signal change at days 0, 3 and 7 post-injection. (**I**) Schematic diagram showing the Annexin-V staining design for live or HK Ctl MSCs (blue cells) or MSC-IPr (red cells). (**J**) Representative flow cytometry analysis of Annexin-V on the surface of live or HK-MSCs. (**K**) Annexin-V MFI of the experiment shown in panels I–J. For panel C, *n* = 6/group with ** *p* < 0.01. For panel E–H, *n* = 5 with * *p* < 0.05 and *** *p* < 0.01. For panel K, *n* = 6/group with * *p* < 0.05 and *** *p* < 0.001. Bars represent SD.

## Data Availability

Reagents generated in this study are available upon reasonable request.

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
