# Peer review of "Humoral Immunity to Allogeneic Immunoproteasome-Expressing Mesenchymal Stromal Cells Requires Efferocytosis by Endogenous Phagocytes"

_cells, 2022, doi:10.3390/cells11040596_

Round 1

Reviewer 1 Report

I reviewed the article “Humoral immunity to allogeneic immunoproteasome-expressing mesenchymal stromal cells requires cross-priming of endogenous phagocytes” by Jean-Pierre Bikorimana, et al. The authors investigate their ability to stimulate humoral immunity.

I have the following comments:

  1. It’s very interesting that MSCs could be converted into potent APCs following some bioengineering means. The authors further investigated OVA-pulsed MSC-IPr cell’s ability to stimulate humoral immunity, and found they elicits a moderate antibody titer, which was further enhanced with the combined use of pro-inflammatory cytokines.
  2. Importantly, allogeneic MSC-IPr cells not only excel at antigen cross-presentation, but can also elicit powerful anti-tumoral responses in vivo, and these imply that these cells could be exploited as cellular vaccines to induce both humoral and cytotoxic immunity.
  3. Lasty, antibody-mediated neutralization of CD47, a potent "do not eat me signal", could enhance antibody titer levels. However, the exact mechanism of phagocytic cells mediating MSC-IPr efferocytosis is unclear; and moreover, in terms of the overall research process and the description of the results, research work of the efferocytosis displayed relatively small in the manuscript. Thus, I suggest the Title be enriched more.

Author Response

First of all, we would like to thank the reviewer for his encouraging and highly constructive comments. As for the raised concerns, the major aim of the study was to investigate whether MSC-IPr cells can mount a humoral response with some side objectives related to their in vivo mode of action. Interestingly, the CD47 "don’t eat me" signal came-out as an interesting modulating effector molecule in this context. Although we find this observation highly interesting, we are currently planning a follow-up study to investigate more in depth the exact involvement of CD47 and other potential efferocytosis-driving signals expressed or secreted by MSCs and recognized by endogenous phagocytic cells. Although we prefer to keep the title as is without inferring more about one mechanism over another to avoid bringing misconceptions or false interpretations/conclusions, we changed the word "cross-priming" by "efferocytosis".   

Reviewer 2 Report

I read with great interest this manuscript by Dr. Bikorimana and colleagues studying a new approach by which MSCs can be converted into efficient antigen-presenting cells. This work clearly shows that de novo expression of the immunoproteasome in MSCs enables them to process/present the antigen to T cells (CD4+). This work adds new knowledge that can help in pharmacological strategies to inhibit tumor growth/metastasis and chemoresistance.

The topic is interesting and well presented. The title is informative, methods well reported and conclusions are consistent with results.

The study is well designed and written and worth publishing.

Author Response

We are delighted and thankful to the great interest and encouragement provided by reviewer 2. 

Reviewer 3 Report

Humoral immunity to allogeneic immunoproteasome-expressing mesenchymal stromal cells requires cross-priming of endogenous phagocytes

In this paper, Bikorimana and colleagues attempt to answer the question as to whether mesenchymal stromal cells (MSC) overexpressing the three inducible proteasome subunits β1i, β2i and β5i (MSC-IPr) represent a potent source of antigens for inducing primary immune responses. Here, they show that injection of MSC-IPr/OVA in mice results in the induction of strong humoral responses with generation of high antibody titers (Fig. 2) directed against I-Ab/OVA peptide complexes and OVA full-length proteins (Fig 4). The ability to these antibodies to efficiently recognize I-Ab/OVA peptide complexes also proved relevant to combat OVA-expressing EG.7 I-Ab tumors in vivo (Fig. 5). Finally, the authors show that the potency of MSC-IPr to prime antibody responses essentially relied on their uptake by phagocytes (Fig. 6).

Altogether, this is a well-presented and -written paper which suggests a potential and unexpected role of immunoproteasomes in donor/apoptotic cells for MHC class II antigen presentation. This manuscript in its present form, however, is not suitable for publication, as discussed below.

Major points

General comment: what would be the beneficial role of immunoproteasomes in MHC class II antigen presentation? As far as I can tell, immunoproteasomes are only relevant in MHC class I antigen processing. How do immunoproteasomes here facilitate the supply of MHC class II-restricted peptides? This point should be clarified.

General comment: the authors refer to “cross-priming” throughout their manuscript, while I see here only analyses of “conventional” MHC class II antigen presentation by phagocytes following ingestion of extracellular material for subsequent presentation to CD4+ T cells. This point also requires clarification.

General comment: it is long assumed that stable proteins are better antigen sources for cross-presentation to CTL than short-lived ones. What is the impact of immunoproteasomes on OVA protein turnover and would an impact on protein half-life in donor/apoptotic cells be relevant to MHC class II antigen presentation?

Fig. 1: The key question of this work is how immunoproteasome overexpression promotes I-Ab upregulation in MSC? MHC class I and II molecules are usually induced by type I and/or II interferon (IFN). Do MSC-IPr exhibit a type I IFN gene signature? And if yes, what would be the cause-and-effect relationship between the acquisition of the inducible subunits and type I IFN production?

Fig. 1: What is the proteasome composition in control MSCs? Are these cells really devoid of immunoproteasome subunits? Are the inducible subunits efficiently incorporated into 26S proteasomes after overexpression? This point could be easily addressed by monitoring the processing of β1i, β2i and β5i by western-blotting.

Fig. 2: Do control MSC fail to induce antibody responses? Or are these lower than that detected following immunization with MSC-IPr? I do not see any control in this experiment.

Fig. 6: Are MSC-IPr more prone to apoptosis than their wild part counterparts? Are apoptotic MSC-IPr more efficiently taken up by phagocytes than control MSC? These points hould be addressed.

Author Response

Comment #1 

What would be the beneficial role of immunoproteasomes in MHC class II antigen presentation? As far as I can tell, immunoproteasomes are only relevant in MHC class I antigen processing. How do immunoproteasomes here facilitate the supply of MHC class II-restricted peptides? This point should be clarified.

Response to comment #1

We thank the reviewer for this comment. In fact, this is indeed an interesting question as the IPr complex is known to be specifically involved in the generation of MHCI-specific peptides with no expected effect on MHCII-mediated antigen presentation. Having said that, we have to remember that the IPr is not strictly involved in antigen processing. We have shown for instance that introduction of the IPr subunits in mesenchymal stromal cells (which only express the constitutive proteasome subunits) leads to dramatic changes in various cellular functions (Abusarah et al. Cell Reports Medicine 2021). These include changes in endosome recycling, autophagy, antigen processing by both MHCI and MHCII, protein homeostasis, the unfolded protein response and metabolism amongst various other processes. It is currently difficult to exactly explain how the IPr triggers MHCII on the surface of murine MSCs in the absence of type I or type II interferon production. Although no CIITA expression could be detected in our transcriptomic analysis, our hypothesis is mostly centered on mitochondrial-related activity. In other words, our published study described in detail enhanced mitochondrial function in MSC-IPr (increased oxidative phosphorylation - Abusarah et al. Cell Reports Medicine 2021). The observed increase in TCA cycle intermediates (e.g. alpha-ketoglutarate, citrate or succinate for example), could lead to important epigenetic changes, which may result in unexpected MHCII-related gene expression - this hypothesis of TCA cycle intermediates could explain the dramatic change in gene expression observed in MSC-IPr cells compared to control MSCs. In addition, MSC-IPr may exhibit autophagy function due to enhanced ROS production by the increased mitochondrial activity (Abusarah et al. Cell Reports Medicine 2021). Since autophagy structures can fuse to recycling endosomes (where we believe IPr-driven peptide generation is taking place in MSC-IPr), 18-24 amino-acid long peptides loading on MHCII and MHCI can occur in parallel in the same endosomal structure. All of the above-mentioned points remain hypothetical open questions and will serve as the basis to several follow-up investigations in that regard.      

Comment #2 

The authors refer to “cross-priming” throughout their manuscript, while I see here only analyses of “conventional” MHC class II antigen presentation by phagocytes following ingestion of extracellular material for subsequent presentation to CD4+ T cells. This point also requires clarification.

Response to comment #2

This is an excellent point and we apologise for the confusion or misunderstanding. In fact, our in vitro data clearly demonstrate that MSC-IPr can promote MHCII-mediated antigen presentation to responding OT-II-derived CD4 T cells. However, the observations made in vivo using allogeneic MSC-IPr cells demonstrate three important points. First, clodronate administration to deplete phagocytes (mostly monocytes/macrophage) delays the in vivo clearance of MSC-IPr cells following their IP infusion. Second, clodronate-treated animals that were vaccinated with MSC-IPr develop low OVA-specific antibody titer, suggesting another cell at play in driving the humoral response. Third, by neutralizing the CD47 "don’t eat me" signal on the surface of MSC-IPr cells, the antibody titer was significantly enhanced. The sum of these attributes clearly indicate that the humoral response in vivo is mostly relying on the transfer of MHCII-generated peptides or any remaining intracellular OVA cargo to endogenous phagocytes. In addition, MSC-IPr cells have been shown to exhibit a strong Th1 cytokine profile highlighted by IL-12 production (Abusarah et al. Cell Reports Medicine 2021). This may explain why OVA-pulsed MSC-IPr administered to mice devoid of phagocytes are limited for their direct ability to trigger a humoral (Th2) response as IL-12 may inhibit this type of immune response.     

Comment #3

It is long assumed that stable proteins are better antigen sources for cross-presentation to CTL than short-lived ones. What is the impact of immunoproteasomes on OVA protein turnover and would an impact on protein half-life in donor/apoptotic cells be relevant to MHC class II antigen presentation?

Response to comment #3

This is indeed a very interesting point. We perfectly agree with the reviewer that protein half-life is extremely important for cross-presentation. However, I must add that the turnover rate of MHC:peptide complex is also extremely important for efficient induction of CTLs. This point was indeed investigated in our previously published study (Abusarah et al. Cell Reports Medicine 2021) where we clearly show that the SIINFEKL:MHCI complex (detected by a monoclonal antibody specific to that complex) is more stable on the surface of MSC-IPr than those presented by monocyte-derived dendritic cells. We have even demonstrated increased antigen presentation when MSC-IPr were treated with chloroquine (which would delay/inhibit endosome acidification resulting in improved antigen preservation overtime). We thus believe that the half-life of both antigen and MHC:peptide complexes is important for efficient CTL induction. As for MHCII presentation, studies related to the detection of MHCII:peptide complexes were not conducted as there is no antibody capable of recognising such complexes so far. However, we know that OVA is efficiently processed by MSC-IPr as experiments conducted using OVA-DQ, a protein emitting fluorescent signals upon degradation/processing, show augmented signal overtime clearly indicating efficient OVA processing. Thus, we can clearly say that MSC-IPr degrades captured OVA very efficiently in the absence of any form of cell death. 

Comment #4 

Fig. 1: The key question of this work is how immunoproteasome overexpression promotes I-Ab upregulation in MSC? MHC class I and II molecules are usually induced by type I and/or II interferon (IFN). Does MSC-IPr exhibit a type I IFN gene signature? And if yes, what would be the cause-and-effect relationship between the acquisition of the inducible subunits and type I IFN production?

Response to comment #4

Part of the answer to this interesting comment can be found in the response provided to comment #1. As for the to type I interferon related comment, we detected no type I IFN gene signature in our MSC-IPr-related studies. Having said that, we are currently investigating the effect of type I IFN (IFNα and IFNβ) on MSCs as these interferons have been reported to reverse the immune-suppressive profile of MSCs by inhibiting NO synthase expression and by inducing STA1-STAT2 dimerization (P. Shou et al. Oncogene 2016). We will, in addition, investigate whether type I IFN modulates antigen cross-presentation and/or the ability of MSCs to express the IPr subunits.    

Comment #5

Fig. 1: What is the proteasome composition in control MSCs? Are these cells really devoid of immunoproteasome subunits? Are the inducible subunits efficiently incorporated into 26S proteasomes after overexpression? This point could be easily addressed by monitoring the processing of β1i, β2i and β5i by western-blotting.

Response to comment #5

In fact, we have already characterized the expression of the IPr subunits in both control MSCs and MSC-IPr by western blotting (Abusarah et al. Cell Reports Medicine 2021 - supplementary files). We clearly demonstrated that control MSCs only express the constitutive proteasome subunits (β1, β2, and β5) with no detectable β1i, β2i and β5i proteins. MSC-IPr, on the other hand, expressed all three IPr subunits. Although the molecular weights for all tested IPr subunits were slightly higher in MSC-IPr cells compared to our positive control (dendritic cells), it is difficult to exclude the notion that such differences could be attributed to variations regarding the glycosylation pattern of these proteins in these two different cell types. As for their incorporation into the 26S proteasome, we are confident that this process has taking place. Otherwise, how can we explain all of the differential effects observed with the MSC-IPr cells?

Comment #6 

Fig. 2: Do control MSC fail to induce antibody responses? Or are these lower than that detected following immunization with MSC-IPr? I do not see any control in this experiment.

Response to comment #6

We have not conducted such experiment for 2 main reasons. First, control MSCs were MHCII negative. Second, we confirmed their inability to activate OT-II derived CD4 T-cell in response to both OVA protein and the OVA-derived MHCII peptide ISQAVHAAHAEINEAGR (data not shown). Nevertheless, we are currently exploring a different line of studies where MSCs are used as a vaccination platform using engineered constructs with various intracellular targeting abilities. This study will include control MSCs (treated with OVA protein or expressing the OVA gene endogenously) as experimental controls. These cellular products will be tested for their ability to trigger both CTLs and humoral responses. 

Comment #7

Fig. 6: Are MSC-IPr more prone to apoptosis than their wild part counterparts? Are apoptotic MSC-IPr more efficiently taken up by phagocytes than control MSC? These points should be addressed.

Response to comment #7

We have been working with the MSC-IPr product for the last 8 years and never observed in vitro cell death. In fact, MSC-IPr survived for 7-10 days after their intraperitoneal infusion in syngeneic C57BL/6 mice, in contrast to shorter times for control MSCs. This implies that MSC-IPr do not undergo spontaneous cell death even after their transfer in syngeneic animals. We must say on the other hand, that thymoproteasome-expressing MSCs express higher level of cell surface phosphatidylserine at steady-states, which explains their enhanced in vivo clearance by efferocytosis (JP. Bikorimana et al. Front. Immunol. 2021). Since such analysis was never conducted on MSC-IPr, we stained both control and MSC-IPr cells with Annexin-V (which binds phosphatidylserine) and found a weaker signal exhibited by MSC-IPr (a new panel added to Figure 6). We can thus conclude that allogeneic MSC-IPr undergo efferocytosis independent of a phosphatidylserine-mediated signal normally used by phagocytes to clear-out apoptotic cells.   

Round 2

Reviewer 3 Report

Humoral Immunity to Allogeneic Immunoproteasome-Expressing Mesenchymal Stromal Cells Requires Efferocytosis by Endogenous Phagocytes

In their response to my comments, Bikorimana and colleagues explain that the requested proteasome characterization of control MSC and MSC-IPr was already performed and described in a previous publication. In this work by Abusarah et al. (PMID: 35028603), none of the three overexpressed β1i, β2i and β5i subunit are processed, as evidenced by their migration profiles in SDS-PAGE (Suppl. Fig. 1B). Indeed, all three introduced immunoproteasome subunits run a bit larger in MSC-IPr than those endogenously expressed by dendritic cells. The authors claim the observed differences in the subunit size may be due to distinct glycosylation patterns but provide no evidence for this assumption. Quite on the contrary, these data imply that the overexpressed β1i, β2i and β5i subunits still contain their N-terminal propeptide and, as such, are not assembled into mature 20S or 26S proteasome complexes.

While I do not dispute the authors’ findings/observations in the present manuscript, I do not believe that these effects are caused by immunoproteasomes. It is not my intention to reject this work. It is however imperative for the sake of clarity and transparency that the authors acknowledge the fact that the observations they made are likely to be due to non-incorporated immunoproteasome subunits.

Author Response

We thank the reviewer for his/her guidance and thorough review of our previous comments. We agree with the reviewer that we have no evidence to support our glycosylation hypothesis and whether this explains the shift observed in the molecular weight of the three IPr subunits. As such, it is difficult to ascertain that the three IPr subunits are indeed incorporated within the mature 20S or 26S of the proteasomal complex in the MSC-IPr cells. We have therefore changed the final sentence of the conclusion in that regard.